# An anthranilic acid-responsive transcriptional regulator controls the physiology and pathogenicity of *Ralstonia solanacearum*

Shihao Song[1,2], Xiuyun Sun[1], Quan Guo[1], Binbin Cui[1], Yu Zhu[1], Xia Li[1], Jianuan Zhou[2], Lian-Hui Zhang[2], Yinyue Deng[1]*

**1** School of Pharmaceutical Sciences (Shenzhen), Shenzhen Campus of Sun Yat-sen University, Sun Yat-sen University, Shenzhen, China, **2** Integrative Microbiology Research Center, College of Plant Protection, South China Agricultural University, Guangzhou, China

* dengyle@mail.sysu.edu.cn

**Data Availability Statement:** All relevant data are within the manuscript and its Supporting Information files.

## Abstract

Quorum sensing (QS) is widely employed by bacterial cells to control gene expression in a cell density-dependent manner. A previous study revealed that anthranilic acid from *Ralstonia solanacearum* plays a vital role in regulating the physiology and pathogenicity of *R. solanacearum*. We reported here that anthranilic acid controls the important biological functions and virulence of *R. solanacearum* through the receptor protein RaaR, which contains helix-turn-helix (HTH) and LysR substrate binding (LysR_substrate) domains. RaaR regulates the same processes as anthranilic acid, and both are present in various bacterial species. In addition, anthranilic acid-deficient mutant phenotypes were rescued by *in trans* expression of RaaR. Intriguingly, we found that anthranilic acid binds to the LysR_substrate domain of RaaR with high affinity, induces allosteric conformational changes, and then enhances the binding of RaaR to the promoter DNA regions of target genes. These findings indicate that the components of the anthranilic acid signaling system are distinguished from those of the typical QS systems. Together, our work presents a unique and widely conserved signaling system that might be an important new type of cell-to-cell communication system in bacteria.

## Author summary

Bacterial wilt caused by *Ralstonia solanacearum* is one of the most widespread, harmful and destructive plant diseases in the world. Our previous study showed that the pathogenic bacterium *R. solanacearum* uses anthranilic acid to regulate the important biological functions, virulence and the production of quorum sensing signals. Here, we show that RaaR, a transcriptional regulator from *R. solanacearum*, was first identified to regulate the same phenotypes as anthranilic acid. Anthranilic acid binds to the LysR_substrate domain of RaaR and enhances the regulatory activity of RaaR to control the target gene expression, including the QS signal synthase encoding genes, *phcB* and *solI*. Both the anthranilic acid synthase TrpEG and the response regulator RaaR are present in diverse bacteria,

**Funding:** This work was financially supported by the Science, Technology and Innovation Commission of Shenzhen Municipality (No. JCYJ20200109142416497 to YD), the National Key Research and Development Program of China (2021YFA0717003 to YD) and Guangdong Forestry Science and Technology Innovation Project (2018KJCX009 to LHZ, 2020KJCX009 to LHZ). The funders had no role in study design, data collection and analysis, decision to publish, or preparation of the manuscript.

**Competing interests:** The authors have declared that no competing interests exist.

suggesting that the anthranilic acid-type signaling system is widespread. Together, our work describes a system where a pathogen uses a single protein to control the bacterial physiology and pathogenesis by responding to anthranilic acid.

## Introduction

Quorum sensing (QS) is a cell-cell communication mechanism used by many bacteria to coordinate group behaviors in response to cell density [1–3]. When the cell density of the microbial population in the environment reaches a threshold, the QS signal activates a responsive regulator to control target gene expression and regulate physiological characteristics. To date, a variety of QS signaling molecules have been discovered, and the first well-characterized QS signaling molecules used by many gram-negative bacteria are the N-acyl homoserine lactones (AHLs) [2,4–15]. As QS system usually regulates the important biological functions and virulence of pathogenic bacteria, many studies have demonstrated that interfering with the QS system is a new strategy to prevent the infection by pathogenic bacteria [16–19].

Bacterial wilt caused by thousands of genetically distinct strains within the *Ralstonia solanacearum* species complex (RSSC) is one of the most important plant diseases in tropical, subtropical, and some warm regions [20]. *R. solanacearum* can infect hundreds of plants belonging to 54 families, causing very large economic losses to global agricultural production [21,22]. There are two QS systems that were previously identified in *R. solanacearum*, the *sol* system and the *phc* system. The *phc* QS system plays a vital role in the regulation of virulence factors, including extracellular polysaccharides (EPS), cell wall-degrading enzymes, and motility [4,21,23–25]. A methyltransferase of *R. solanacearum* named PhcB synthesizes either methyl 3-hydroxymyristate (3-OH MAME) or methyl 3-hydroxypalmitate (3-OH PAME) as the signal of the *phc* QS system [23]. The signals phosphorylate the response regulator PhcR by activating the histidine kinase PhcS and then finally control the target genes and the *sol* system [4,26–28]. The *sol* system is a typical QS system based on acyl-homoserine lactone (AHL) signals, but does not regulate virulence factor production in *R. solanacearum* AW1 strain [4,23,28].

Our previous study showed that *R. solanacearum* GMI1000 utilizes anthranilic acid to regulate important biological functions and the synthesis of QS signals, including 3-OH MAME and AHL signals [29]. In this study, we identified a receptor responsive to anthranilic acid, RaaR (Regulator of anthranilic acid of *Ralstonia solanacearum*). RaaR is a transcriptional regulator with two domains, an HTH domain and a LysR_substrate domain. Perception of anthranilic acid by the LysR_substrate domain enhances the binding of RaaR to the target gene promoter and then controls the physiology and pathogenicity of *R. solanacearum*. We also found that homologs of both the anthranilic acid synthase TrpEG and the regulator RaaR are present in diverse bacteria, suggesting that the anthranilic acid-type signaling system is widespread.

## Results

### RaaR is a downstream component of the anthranilic acid signaling system in *R. solanacearum*

Our previous studies showed that anthranilic acid regulates the important biological functions and virulence of *R. solanacearum* [29]. As anthranilic acid is a precursor for the biosynthesis of 4-hydroxy-2-heptylquinoline (HHQ) and the *Pseudomonas* quinolone signal (PQS) in

*Pseudomonas aeruginosa*, we then searched for homologs of MvfR, which is the receptor of both HHQ and PQS signals in *P. aeruginosa* [30–32], in the genome of *R. solanacearum* GMI1000 to identify the potential downstream component of the anthranilic acid signaling system by using the BLAST program (https://blast.ncbi.nlm.nih.gov/Blast.cgi). We found 10 potential homologs of MvfR sharing 15.18% to 33.5% identity with MvfR (S1 Table). We *in trans* expressed each of all the homologs in the anthranilic acid deficient mutant Δ*trpEG* and measured the motility activity of these overexpression strains. The results showed that only *in trans* expression of *RSp0912* rescued the defective motility phenotype of the Δ*trpEG* strain (Fig 1A and 1B). Then, *RSp0912* was selected for further investigation. We found that *in trans* expression of *RSp0912* restored all the tested phenotypes of Δ*trpEG* to the wild-type strain levels, including biofilm formation, EPS production and cellulase production, which are all regulated by the anthranilic acid signaling system (Fig 1C, 1D and 1E). We then measured the gene expression levels of *RSp0912* in the wild-type strain and the *trpEG* deletion mutant. The results indicated that deletion of *trpEG* caused a significant decrease in the gene expression level of *RSp0912*, and exogenous addition of anthranilic acid could almost restore the expression level of *RSp0912* (S1 Fig). As RSp0912 has a helix-turn-helix (HTH) domain and a LysR_substrate domain (Fig 1B), we hypothesized that it might be a downstream component of the anthranilic acid signaling system and thus named it the *r*egulator of *a*nthranilic *a*cid of *Ralstonia solanacearum* (RaaR).

## RaaR is a key regulator of the anthranilic acid signaling system

As *in trans* expression of *raaR* fully restored the defective phenotypes of the anthranilic acid deficient mutant Δ*trpEG*, we then continued to investigate the role of RaaR in regulating these important biological functions. An in-frame deletion mutant of *raaR* was generated, we found that deletion of *raaR* caused the same phenotypic changes as observed with the anthranilic acid mutant Δ*trpEG*, including reduced motility activity (Fig 2A), biofilm formation (Fig 2B), EPS production (Fig 2C), and cellulase production (Fig 2D), but did not affect the growth rate of the bacterial cells in either nutrient-rich or nutrient-poor medium (S2 Fig). And the addition of exogenous anthranilic acid didn't restore these phenotypes of both the *raaR* deletion mutant and the *trpEG* and *raaR* double deletion mutant to wild-type strain levels (Fig 2). From this, we concluded that RaaR is a key downstream regulator of the anthranilic acid signaling system in *R. solanacearum*.

## RaaR regulates target gene expression by directly binding to promoters

Our previous study demonstrated that anthranilic acid positively regulates both the *phc* and *sol* QS systems in *R. solanacearum* [29]. We then studied whether RaaR also regulates the QS systems. We constructed P*phcB-lacZ* and P*solI-lacZ* reporter systems in a *raaR* mutant and a *trpEG* and *raaR* double deletion mutant, as both of the target genes are positively controlled by anthranilic acid [29]. Deletion of *raaR* resulted in reduced expression levels of both *phcB* and *solI* (Fig 3A and 3B), which are the signal molecule synthase-encoding genes of the *phc* and *sol* QS systems, respectively. Expression levels of both *phcB* and *solI* in the *trpEG* and *raaR* double deletion mutant also decreased to *raaR* deletion mutant levels. Therefore, we measured and compared the production of 3-OH MAME, C8-AHL and C10-AHL in the wild-type, *raaR* mutant, *trpEG* and *raaR* double deletion mutant and complemented strains. The production of 3-OH MAME, C8-AHL and C10-AHL was reduced in the *raaR* mutant strain and the *trpEG* and *raaR* double deletion mutant strain, and *in trans* expression of *raaR* almost fully rescued signal production (Fig 3C).

RaaR is a key downstream regulator of the anthranilic acid signaling system and contains an HTH domain that is predicted to be closely related to DNA binding. Therefore, we tested

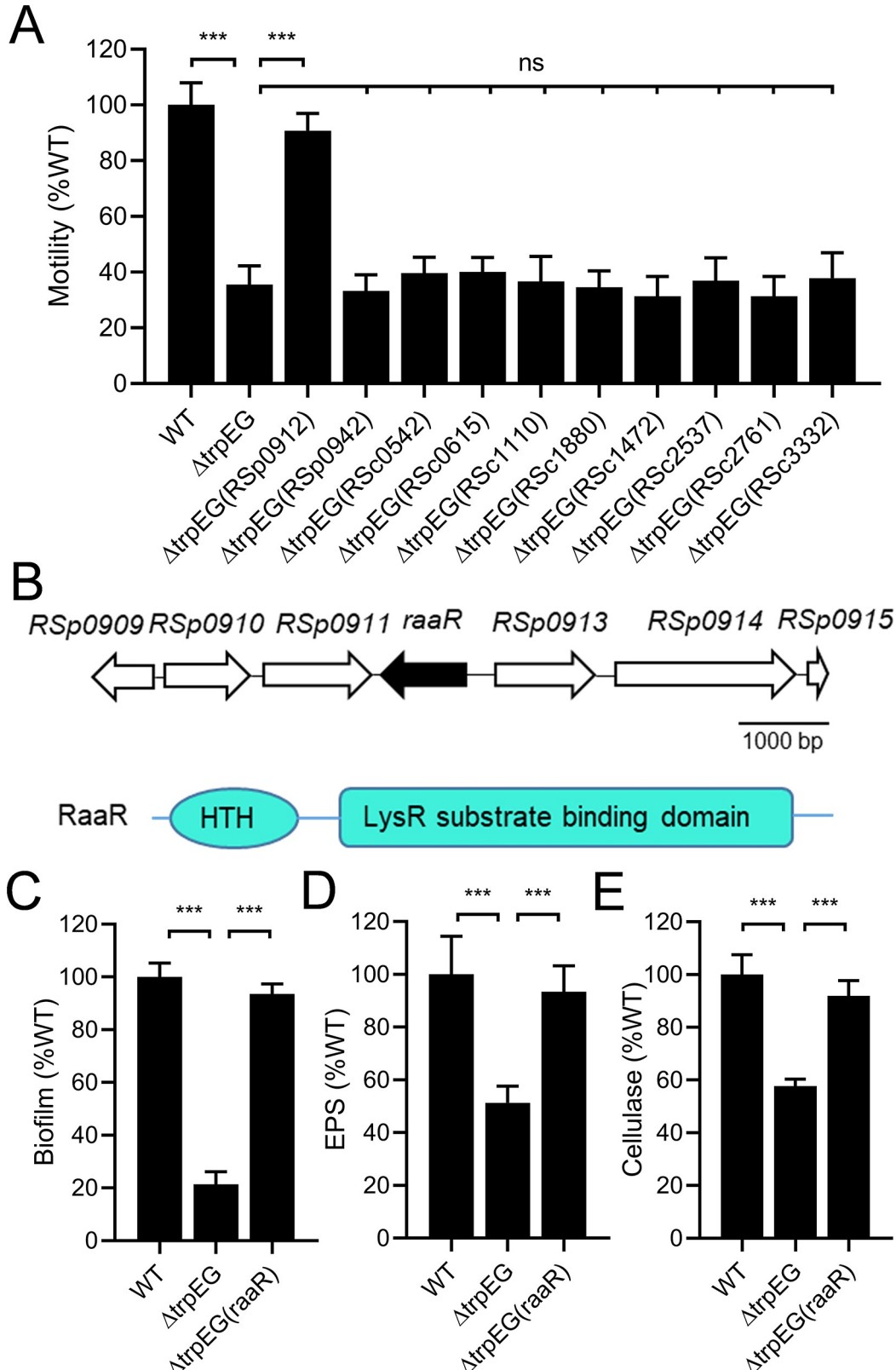

**Fig 1. Influence of *raaR* on the virulence-related phenotypes in the *trpEG* mutant strains.** (A) Effects of the homologs of MvfR in *R. solanacearum* on the motility activity of the *trpEG*-deficient mutant of *R. solanacearum* GMI1000. (B) Genomic

organization of the *raaR* region in *R. solanacearum* GMI1000 (top). Domain structure analysis of RaaR (bottom). The RaaR protein has two domains: an HTH domain and a LysR_substrate domain. *In trans* expression of RaaR restored biofilm formation (C), EPS production (D) and cellulase production (E) in the TrpEG-deficient mutant. The data are the means ± standard deviations of three independent experiments. ***$p < 0.001$ (unpaired *t*-test).

whether the transcriptional regulation of *phcB* and *solI* is achieved by direct binding of RaaR to their promoters by performing electrophoretic mobility shift assays (EMSAs). A 327-bp DNA fragment from the *phcB* promoter and a 324-bp DNA fragment from the *solI* promoter were PCR-amplified and used as probes. As shown in Fig 3, the *phcB* and *solI* promoter DNA fragments formed stable DNA-protein complexes with RaaR and migrated slower than unbound probes. The amount of labeled probe that bound to RaaR increased with increasing amounts of RaaR but decreased in the presence of a 100-fold greater concentration of the unlabeled probe (Fig 3D and 3E). We also selected a housekeeping gene *gyrB*, which encodes DNA topoisomerase (ATP-hydrolyzing) subunit B, and used it as a negative control. Deletion of *raaR* didn't affect the expression level of *gyrB*, and RaaR didn't bind to the *gyrB* promoter DNA (S3 Fig).

## Anthranilic acid is a signal ligand of RaaR

Since RaaR regulates the same processes as anthranilic acid signal and has a LysR_substrate domain, we hypothesized that anthranilic acid may influence the activity of RaaR through ligand-protein interactions [33]. To confirm this hypothesis, we performed isothermal titration calorimetry (ITC) analysis to test whether RaaR binds anthranilic acid. RaaR, which contained 313 aa with a calculated molecular weight of 35.59 kDa, was purified to homogeneity using affinity chromatography and characterized for interactions with anthranilic acid. Anthranilic acid binds strongly to the purified RaaR protein (Fig 4). Fitting the binding isotherm data showed that RaaR binds to anthranilic acid in a 1:1 stoichiometry with an estimated dissociation constant (Kd) of 3.63 μM. In addition, we purified a polypeptide containing only the LysR_substrate domain of RaaR and tested it for anthranilic acid binding. ITC analyses revealed that the LysR_substrate domain is the anthranilic acid binding motif (S4 Fig).

To study how anthranilic acid influences the binding of RaaR to promoter DNA, we used circular dichroism (CD) spectroscopy to study anthranilic acid-RaaR interactions and detect secondary structural changes in the RaaR protein when it bound anthranilic acid. The CD spectrum of α-helixes is characterized by two negative bands of almost equal intensities at 222

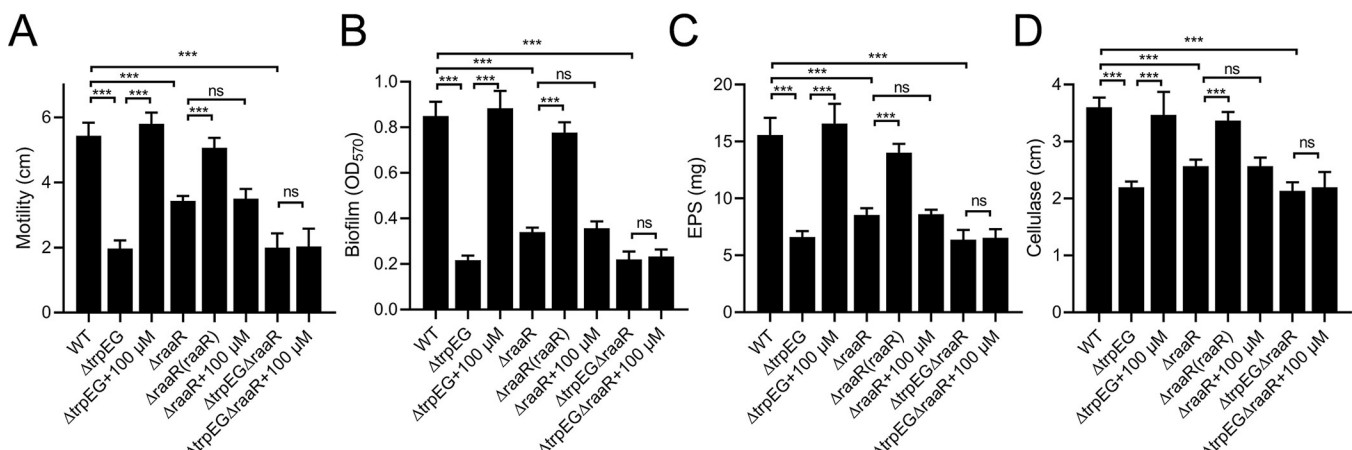

**Fig 2.** Effects of *raaR* on the anthranilic acid-regulated motility (A), biofilm formation (B), EPS production (C) and cellulase production (D) in *R. solanacearum* GMI1000. The data are the means ± standard deviations of three independent experiments. ***$p < 0.001$ (unpaired *t*-test).

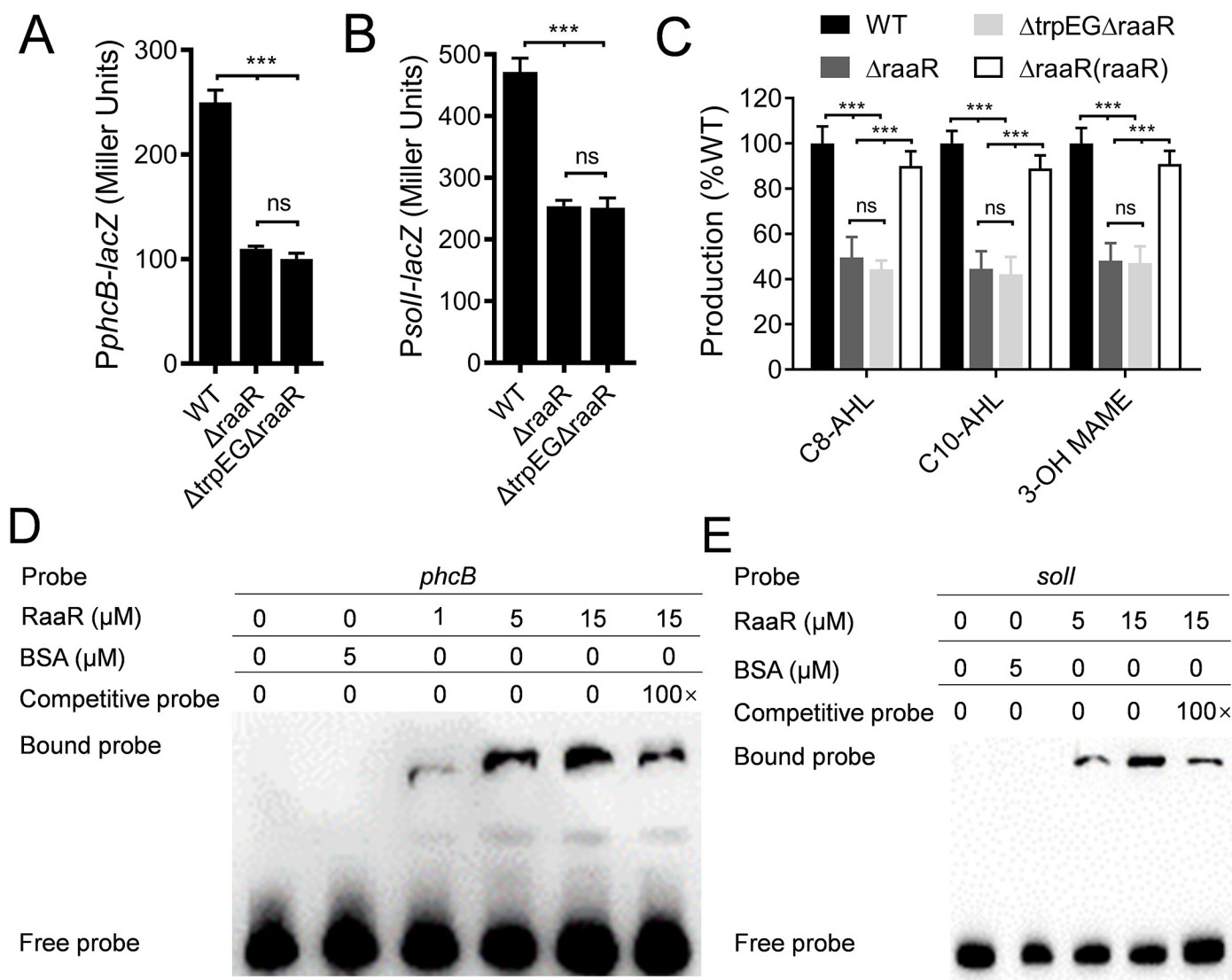

**Fig 3. Influence of *raaR* on the QS signal production of *R. solanacearum*.** Effects of *raaR* on the gene expression levels of *phcB* (A) and *solI* (B) and on QS signal production (C) in *R. solanacearum* GMI1000. We used promoter activity assays to quantify gene expression. The data are the means ± standard deviations of three independent experiments. ***$p < 0.001$ (unpaired t-test). EMSA analysis of the *in vitro* binding of RaaR to the promoters of *phcB* (D) and *solI* (E); biotin-labeled 327-bp *phcB* and 324-bp *solI* promoter DNA probes were used for the protein binding assay. A protein-DNA complex, represented by a band shift, was formed when different concentrations of RaaR protein were incubated with the probe at room temperature for 30 min.

and 208 nm along with a strong positive band at approximately 192 nm. The β-sheet spectrum is characterized by a negative band at 216 nm along with a strong positive band at approximately 195 nm [34]. As shown in S5 Fig, both the α-helix and β-sheet spectra of RaaR significantly changed when it was supplemented with 10 μM anthranilic acid (S5 Fig). These results are consistent with the ITC results, indicating that the RaaR protein binds to anthranilic acid signals and the binding significantly changes the secondary structure of the RaaR protein.

## Anthranilic acid enhances RaaR binding to target promoter DNA

To determine how the binding of anthranilic acid to RaaR might affect the activity of RaaR, we then examined the effects of anthranilic acid on the binding of RaaR to the *phcB* and *solI*

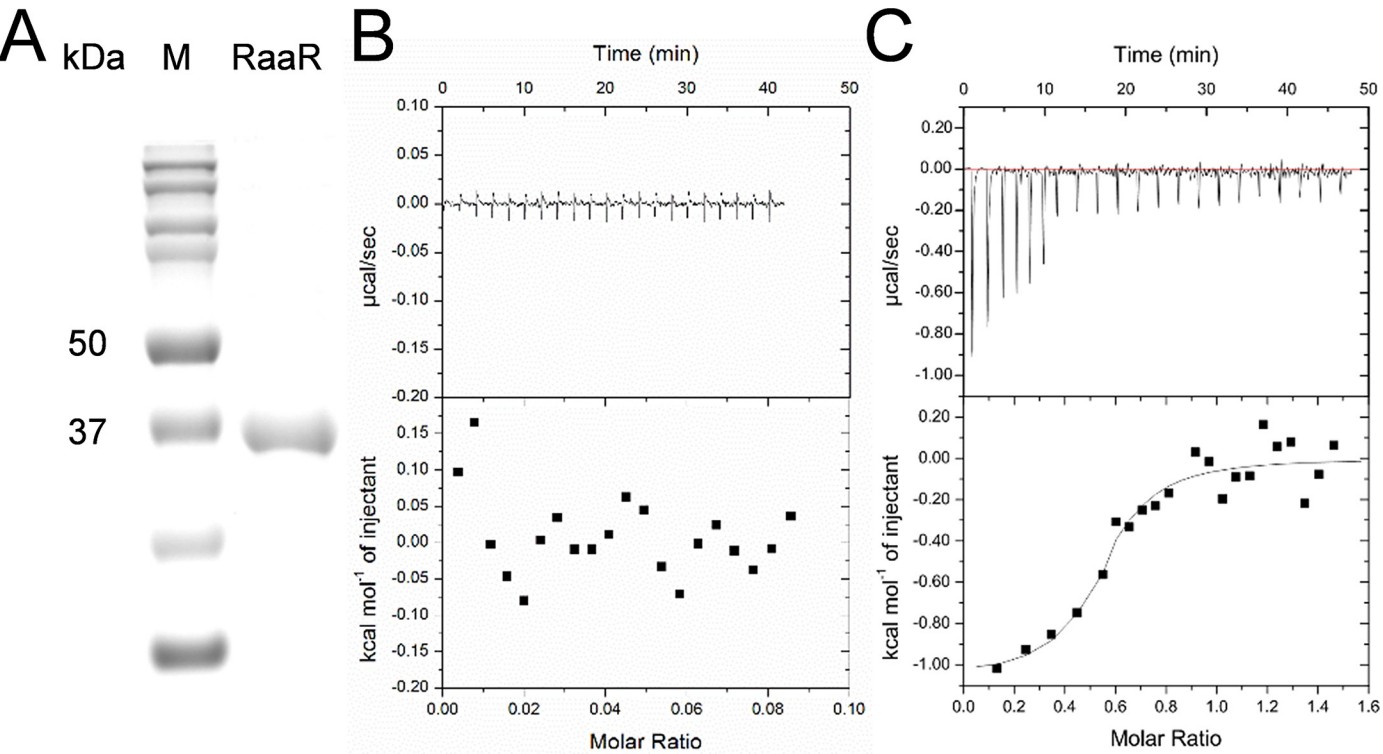

**Fig 4. ITC analysis of the binding of anthranilic acid to RaaR protein.** (A) SDS-PAGE of the purified RaaR protein. (B) ITC titration of 250 μM anthranilic acid in PBS buffer at 25˚C. (C) ITC titration of 20 μM RaaR with 250 μM anthranilic acid in PBS buffer at 25˚C.

promoters by EMSA. As shown in Fig 5, the binding of RaaR to the *phcB* and *solI* promoter probes was enhanced when anthranilic acid was present in the reaction mixtures. The amounts of the probes that bound to RaaR increased with increasing amounts of anthranilic acid, suggesting that the addition of anthranilic acid increased the ability of RaaR to bind to the target promoter DNA (Fig 5). To further determine whether the effect of anthranilic acid on RaaR

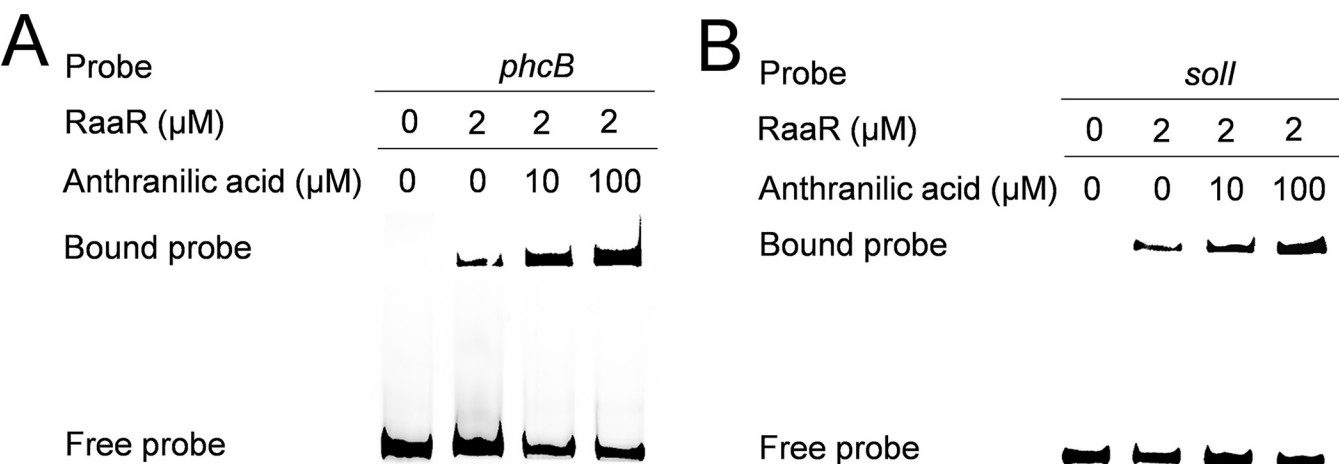

**Fig 5.** The effects of anthranilic acid on the binding of RaaR to the promoters of *phcB* (A) and *solI* (B) were assessed by performing EMSA *in vitro*. A protein-DNA complex was formed when the protein was incubated with the probe, and different concentrations of anthranilic acid had an effect on the formation of the complex at room temperature for 30 min.

was specific. We evaluated the effect of one of the analogs of anthranilic acid, HHQ, on the ability of RaaR to bind to the target promoter DNA. The results showed that HHQ did not affect the RaaR binding to the target promoter DNA even at a final concentration of 100 μM (S6 Fig).

## RaaR contributes to *R. solanacearum* pathogenicity

Previous studies characterized the attenuated virulence of an anthranilic acid synthase-negative mutant of *R. solanacearum* [29]. We evaluated the effect of RaaR on the ability of *R. solanacearum* to infect host plants. As shown in Fig 6A, compared with the plants infected by the *R. solanacearum* wild-type strain and complemented strains, tomato plants infected by the *R. solanacearum raaR* mutant strain were significantly reduced in wilting symptoms. After 14 days of inoculation, the disease indexes of tomato plants inoculated with wild-type and complemented strains of *R. solanacearum* were 3.7 and 3.5, respectively, while that of Δ*raaR* was 1.8, which was 51.4% lower than that of the wild-type strain (Fig 6B).

We further quantified the colony-forming units (CFU) of *R. solanacearum* strains in both the roots and stems of the tomato plants. The numbers of CFUs recorded for the *R. solanacearum* wild-type, Δ*raaR* mutant, and complemented strains were $4.27 \times 10^8$, $0.53 \times 10^8$, and $3.73 \times 10^8$ per gram of root tissue at 7 d postinoculation, respectively (Fig 6C). A similar result was observed for tomato stems, in which the numbers of CFUs of the three strains were $6.83 \times 10^8$, $0.91 \times 10^8$, and $6.03 \times 10^8$ per gram of stem tissue at 7 d postinoculation, respectively (Fig 6D). These results suggest that RaaR plays an important role in the pathogenesis of *R. solanacearum*.

## RaaR controls the expression of a wide range of genes

To further study the regulatory roles of RaaR in controlling bacterial physiology, we analyzed and compared the transcriptomes of the wild-type strain and the Δ*trpEG* and Δ*raaR* mutants by using RNA sequencing (RNA-Seq). Differential gene expression analysis showed that 41 genes were increased and 145 genes were decreased in the Δ*raaR* mutant compared with their expression in the wild-type strain (Log$_2$ fold-change ≥1.5), whereas 227 genes were increased and 278 genes were decreased in the Δ*trpEG* mutant (S7 Fig and S2 Table). These differentially expressed genes are associated with a range of biological functions, including motility, virulence, regulation, transport and signal transduction (S7 Fig and S2 Table). We also compared the transcriptome profiles of the Δ*trpEG* mutant and the Δ*raaR* mutant and found an overlap in their target genes (S7B and S7C Fig).

## The anthranilic acid signaling system is unique and widespread in bacteria

In *P. aeruginosa*, anthranilic acid is a precursor of the QS signal PQS. The PQS signal is sensed by the receptor protein MvfR to regulate the biological functions and virulence of *P. aeruginosa* [15,30–32]. To verify whether the *mvfR* gene of *P. aeruginosa* could functionally replace *raaR*, the coding region of *mvfR* was cloned in an expression vector and conjugated to the *raaR* deletion mutant. As shown in S8 Fig, the expression of *mvfR* in the Δ*raaR* mutant does not restore the phenotypic changes, indicating that *mvfR* is not a functional homolog of *raaR*. We then expressed and purified the MvfR protein by affinity chromatography and examined its interaction with anthranilic acid by ITC. The results showed that the MvfR protein did not bind to anthranilic acid (S9 Fig). We also found that the RaaR protein did not bind to PQS, HHQ or 2,4-dihydroxyquinoline (DHQ) (S10 Fig). Taken together, these results suggested that anthranilic acid specifically binds to RaaR and that the anthranilic acid signaling system of *R. solanacearum* is completely different from the *pqs* system in *P. aeruginosa*. To investigate whether

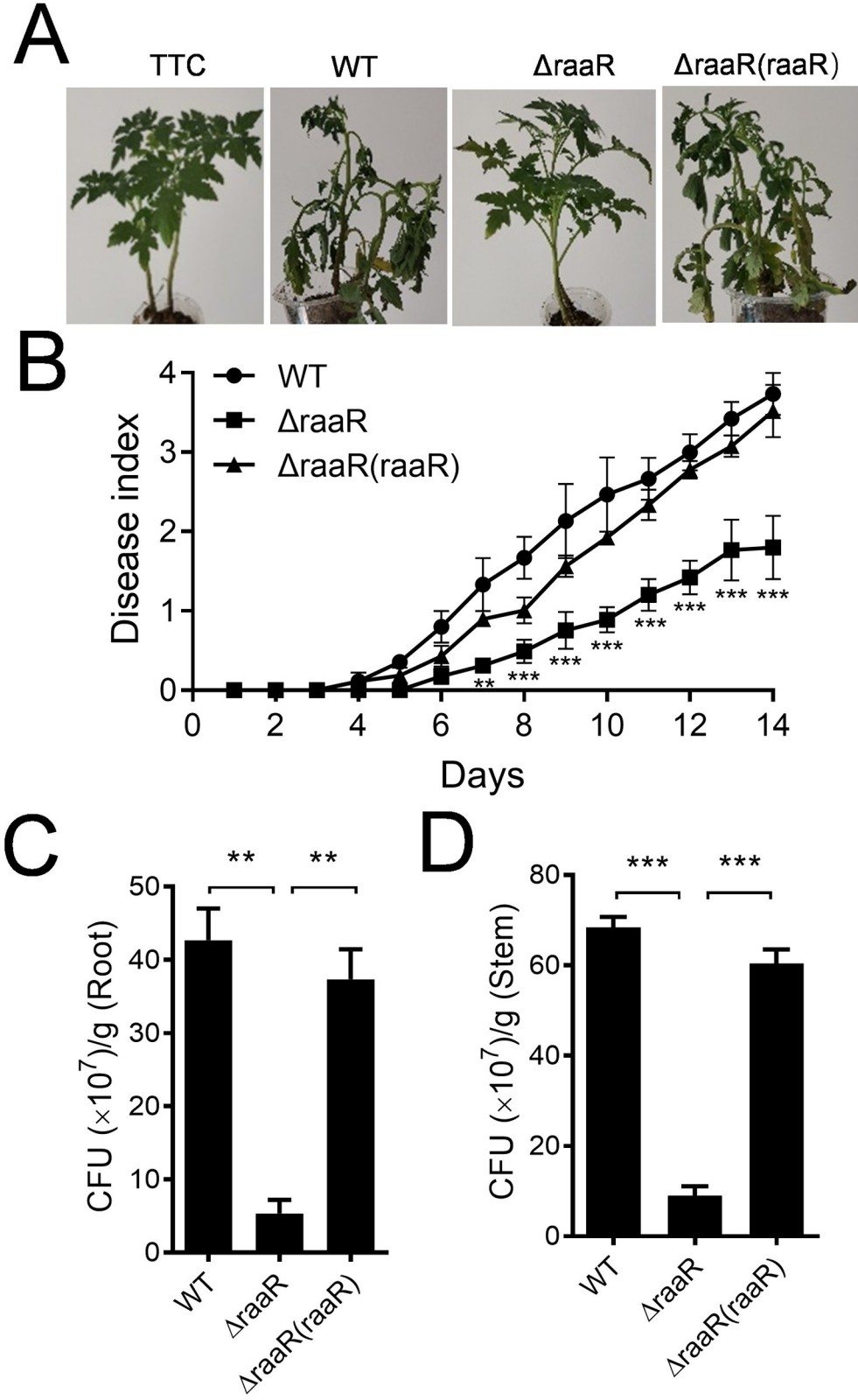

**Fig 6. Influence of *raaR* on the pathogenicity of *R. solanacearum* in tomato plants.** (A) Analysis of *R. solanacearum* virulence in tomato plants at 14 days after inoculation of *R. solanacearum*. (B) The effects of *raaR* on the virulence of *R. solanacearum* were measured by assessing the disease index of bacterial wilt in tomato. ∗∗$p < 0.01$; ∗∗∗$p < 0.001$

(unpaired ANOVA test). The number of CFUs of the *R. solanacearum* wild-type, *raaR* mutant and complemented strains in the roots (C) and stems (D) of tomato plants. The data are the means ± standard deviations of three independent experiments. $^{**}p < 0.01$; $^{***}p < 0.001$ (unpaired *t*-test).

the anthranilic acid signaling system is widely present in bacteria, both *trpEG* and *raaR* homologs were sought in the genome database by BLAST. It was revealed that the anthranilic acid signaling system likely operates in many different bacterial species, including in the bacterial species of *Burkholderiales*, *Caballeronia*, *Trinickia*, *Pandoraea*, and *Cupriavidus* (S3 Table).

From the results of this study, we can build a model for how the anthranilic acid-mediated signaling system controls gene expression in *R. solanacearum*. At high cell densities, when the concentration of anthranilic acid reaches a threshold, it binds to RaaR to cause allosteric conformational changes and enhance the RaaR ability to bind target gene promoters (S11 Fig).

## Discussion

Our previous study found that anthranilic acid controls important biological functions and virulence in *R. solanacearum* [29]. In this study, we identified that the transcriptional regulator RSp0912, which we named RaaR, is a receptor protein of anthranilic acid. RaaR regulates the same processes as anthranilic acid, including motility, biofilm formation, QS signal production and virulence (Figs 2, 3 and 6). In addition, the anthranilic acid mutant phenotypes were rescued by *trans* expression of RaaR (Fig 1). RaaR has an HTH domain and a LysR_substrate domain (Fig 1B). Anthranilic acid was shown to bind to the LysR_substrate domain of RaaR with high affinity and enhanced RaaR binding to target promoter DNA through induction of allosteric conformational changes (Figs 4 and 5, S4 and S5). A previous study showed that 9 site-specific substitution mutations affected MvfR activity [32]. To identify the critical residues in RaaR, we compared RaaR with MvfR using amino acid sequence alignments. Analysis of the two proteins identified four conserved amino acid residues at positions A171, L191, L192 and I249 that might be critical for the interaction between RaaR and anthranilic acid. We then generated four single point mutants (RaaR$^{A171F}$, RaaR$^{L191A}$, RaaR$^{L192A}$ and RaaR$^{I249A}$). ITC analysis showed that mutations at L191 and L192 abolished the binding between RaaR and anthranilic acid (S12 Fig). Our results suggest that RaaR is a specific receptor protein of anthranilic acid and represents a new one-component signaling system.

In *P. aeruginosa*, anthranilic acid is a precursor of the QS signal PQS. The PQS signal is sensed by the receptor protein MvfR to regulate the phenotypes and virulence of *P. aeruginosa* [15,30–32]. Consistent with the results of our previous study, the findings in this study also suggest that the anthranilic acid signaling system is distinguished from the *pqs* QS system in *P. aeruginosa*. The addition of PQS, HHQ or DHQ resulted in no restoration of the anthranilic acid mutant phenotypes, and *trans* expression of *mvfR* showed no effect on the defective phenotypes of the Δ*raaR* mutant (S8 Fig). We also revealed that the MvfR protein binds the PQS signal but not anthranilic acid (S9 Fig). In addition, the RaaR protein does not bind PQS, HHQ or DHQ (S10 Fig). Collectively, our findings in this study support that the anthranilic acid/RaaR system is different from the PQS/MvfR system in *P. aeruginosa*.

Previous studies have demonstrated that the *phc* system regulates the *solIR* system [4,26,28]. Our recent study showed that the expression levels of both *phcB* and *solI* and the production of 3-OH MAME and AHL signals were reduced in the *trpEG* deletion mutant [29]. The findings in this study showed that the anthranilic acid signaling system regulates the transcriptional expression levels of *phcB* and *solI* and the production of AHL and 3-OH MAME signals through the RaaR receptor (Fig 3). We also found that RaaR directly binds to the promoters of *phcB* and *solI* and that exogenous addition of anthranilic acid enhances the binding (Figs 3

and 5). Interestingly, RaaR can also regulate the transcriptional expression levels of *epsA* by directly binding to the promoter of *epsA* (S13 Fig), suggesting a complicated hierarchy of signaling systems in *R. solanacearum*. To further study the binding site of the RaaR protein in the *phcB*, *solI* and *epsA* promoters, we compared and analyzed the promoter sequences of the three genes and identified the potential binding site sequence as GCGGGTGCG. EMSA analysis showed that no DNA–protein complexes formed when the GCGGGTGCG fragment was deleted from the promoter regions of *phcB*, *solI* and *epsA* (S14 Fig), suggesting that this fragment is essential for the binding of RaaR to the *phcB*, *solI* and *epsA* promoters.

In addition, a BLAST search with the TrpEG/RaaR homologs revealed that both the signal synthase gene and sensor gene are widely present in many bacterial species, including in other RSSC phylotypes like *R. solanacearum* RS476 (phylotype I), *R. solanacearum* CFBP2957 (phylotype IIA), *R. solanacearum* UW551 (phylotype IIB), *R. solanacearum* CMR15 (phylotype III) and *R. solanacearum* PSI07 (phylotype VI), *B. cepacia*, *P. piptadeniae*, *C. calidae*, *T. symbiotica*, *P. thiooxydans*, and *C. gilardii* (S3 Table). Intriguingly, some bacteria present only one homolog of TrpEG or RaaR (S3 Table), suggesting that anthranilic acid could be used in both intraspecies signaling and cross-talking, or as a common metabolic product. Together, these findings will trigger further investigation of the roles and mechanisms of this signaling system in diverse bacterial genera.

## Materials and methods

### Bacterial strains and growth conditions

The bacterial strains and plasmids used in this work were listed in S4 Table. *R. solanacearum* GMI1000 (ATCCBAA-1114) was obtained from the American Type Culture Collection (ATCC) and maintained at 28˚C in casamino acid-peptone-glucose (CPG medium) [29,35]. The following antibiotics were added when necessary: ampicillin and kanamycin, 100 μg mL$^{-1}$; tetracycline, 10 μg mL$^{-1}$; and gentamicin, 10 μg mL$^{-1}$. Bacterial growth was determined by measuring the optical density at 600 nm.

### Protein expression and purification

The coding regions of *raaR* and *mvfR* were amplified with the primers listed in S5 Table with *Hin*dIII and *Eco*RI restriction sites and fused to the expression vector pMAL-C5X. The fusion gene constructs were transformed into *E. coli* strain BL21. Affinity purification of RaaR and MvfR proteins was performed following a method described previously [36]. *E. coli* BL21 cells expressing RaaR and MvfR were sonicated in a solution containing 20 mM Na-HEPES pH 8.1, 1M NaCl, and 5 mM DTT (solution A). The cleared lysate was applied to a maltose binding protein (MBP) affinity column. The MBP fusion RaaR or MvfR was eluted by solution A supplemented with 40 mM maltose, and the MBP tag was then cleaved by tobacco etch virus protease. The RaaR and MvfR proteins were pooled and concentrated, then shock-frozen in liquid nitrogen and stored at −80˚C for further assay.

### Electrophoretic mobility shift assay

The DNA probes used for the EMSA were prepared by PCR amplification using the primer pairs listed in S5 Table [37,38]. The purified PCR products were 3-end-labeled with biotin following the manufacturer's instructions (Thermo). The DNA-protein binding reactions were performed according to the manufacturer's instructions (Thermo). A 5% polyacrylamide gel was used to separate the DNA-protein complexes. After UV cross-linking, the biotin-labeled probes were detected in the membrane using a Biotin luminescent detection kit (Thermo).

## ITC and CD spectroscopy analysis

ITC measurements were performed using an ITC-200 microcalorimeter following the manufacturer's protocol (MicroCal, Northampton, MA) [37]. In brief, titrations began with one injection of 2 μL of anthranilic acid (250 μM) solution into a sample cell containing 350 μL of RaaR solution (20 μM) in the MicroCal ITC-200 microcalorimeter. The heat changes accompanying injections were recorded. The titration experiment was repeated at least three times, and the data were calibrated with the final injections and fitted with a one-site model to determine the binding constant (Kd) using MicroCal ORIGIN version 7 software. Circular dichroism (CD) analysis of RaaR was carried out on a Chirascan spectropolarimeter (Applied Photophysics, UK) as previously described [33]. RaaR and anthranilic acid solutions were mixed at room temperature for 1 h at a final concentration of 10 μM.

## Construction of in-frame deletion mutants and complementation

*R. solanacearum* GMI1000 was used as the parental strain for the generation of in-frame deletion mutants following methods described previously [29,39]. The primers used to generate the upstream and downstream regions flanking *raaR* are listed in S5 Table. For complementation analysis, the coding regions of the regulator genes were amplified and cloned into the pBBR1 MCS-2 plasmid. The resulting constructs were introduced into *R. solanacearum* GMI1000 deletion mutants using electroporation.

## Phenotype assays

For analysis of biofilm formation [40], a single colony of each strain was inoculated and grown overnight at 28˚C with agitation in 5 mL of CPG liquid medium. Bacterial cells were inoculated 1:100 in CPG medium in 96-well polystyrene plates. After incubation at 28˚C for 24 h, the cells were stained with 0.1% crystal violet (CV) for 30 min. The planktonic cells were removed by several rinses with $H_2O$. The CV-stained bound cells were air dried for 1 h and then dissolved in 95% ethanol, and the optical density at 570 nm ($OD_{570}$) of the solution was measured to quantify biofilm formation.

Cellulase production was determined on carboxymethylcellulose sodium (CMS) solid medium (1 liter contained 1 g of CMS, 3.8 g of $Na_3PO_4$, and 8.0 g of agar, pH 7.0) [41]. The CMS medium was added to a culture dish (Φ = 9 cm). An overnight culture was diluted to an $OD_{600}$ of approximately 0.1, and 2 μL of the bacterial suspension was inoculated into the center of the CMS plates. The plates were incubated at 28˚C for 48 h. The plates were stained with 0.5% Congo red for 30 min. The plates were rinsed three times with 1 M NaCl. Then, a transparent ring was apparent, and its size was measured.

Swarming motility was determined on semisolid agar (0.3%) [42]. Bacteria were inoculated into the center of plates containing 1% tryptone (Becton, Dickinson and Company, Maryland, USA) and 0.3% agar (Becton, Dickinson and Company, Maryland, USA). The plates were incubated at 28˚C for 48 h before the diameter of the resulting colonies was measured.

For quantification of EPS production, bacteria were inoculated into sucrose and peptone (SP) liquid medium (1 liter contained 5 g of peptone, 20 g of sucrose, 0.5 g of $KH_2PO_4$, and 0.25 g of $MgSO_4$, pH 7.2) [43]. A 100-mL aliquot of the culture ($OD_{600}$ = 3.0) was collected and centrifuged at 12,000 rpm for 20 min. The supernatants were filtered through a 0.22-μm membrane. The collected supernatants were mixed with 4 volumes of absolute ethanol, and the mixture was incubated at 4˚C overnight. The precipitated EPS were isolated by centrifugation and dried overnight at 55˚C before the dry weight was determined.

## Bacterial growth analysis

An overnight bacterial culture in CPG liquid medium was washed twice in fresh CPG liquid medium or MP liquid minimal medium and inoculated to an $OD_{600}$ of 0.01 in fresh CPG medium and of 0.1 in fresh MP minimal medium. A 200-μL cell suspension was grown in each well at 28˚C with low-intensity shaking using the Bioscreen C automated growth curve analysis system. CPG medium or MP minimal medium was used as the negative control. MP minimal medium (1 liter): $FeSO_4 \cdot 7H_2O$, $1.25 \times 10^{-4}$ g; $(NH_4)_2SO_4$, 0.5 g; $MgSO_4 \cdot 7H_2O$, 0.05 g; and $KH_2PO_4$, 3.4 g. The pH was adjusted to 7, and 20 mM glutamate was added [29,44,45].

## Construction of reporter strains and measurement of β-Galactosidase activity

We used promoter activity assays to quantify gene expression. The promoters of *phcB*, *solI* and *raaR* were amplified using the primer pairs listed in S5 Table. The resulting products were inserted upstream of the promoterless *lacZ* gene in the vector pME2-*lacZ*. Transconjugants were then selected on CPG agar plates supplemented with tetracycline and X-gal. Measurement of β-galactosidase activities was performed following previously described methods [19,38].

## LC-MS analysis

*R. solanacearum* cells were cultured in CPG liquid medium for 48 h, and one liter of culture supernatant was collected by centrifugation and extracted with equal volume of ethyl acetate. The culture supernatants were extracted twice with ethyl acetate, the solvent was evaporated, and the residue was dissolved in methanol. Analyses were performed by using a Waters ACQUITY UPLC/Xevo G2 QTOF system (Waters, Milford- Massachusetts, USA), which consisted of an ACQUITY UPLC system and a Waters Q-TOF Premier high-resolution mass spectrometer in negative electrospray ionization mode interfaced with an ACQUITY UPLC BEH C18 column (2.1×50 mm). Elution was performed via a gradient of 5–100% $CH_3OH$ in water supplemented with 0.01% formic acid at a flow rate of 0.4 mL/min for 10 min. Then, 100% $CH_3OH$ was used for 2 min, and 5% $CH_3OH$ was used for 3 min. The entire column eluate was introduced to the Q-TOF mass spectrometer according to the manufacturer's instructions [29].

## Virulence assays in tomato plants

Analysis of the virulence of the *R. solanacearum* WT, Δ*raaR* and complemented strains was performed in an AIRKINS greenhouse (28˚C, light 16 h and dark 8 h). A mixture including field soil, sand, and compost (1.25:1.25:0.5) was prepared and autoclaved at 121˚C for 20 min. Tomato seeds (Jinfeng 1) were surface-sterilized in 2% NaClO for 3 min and 75% ethanol for 2 min, rinsed 3 times in sterile water, and then planted in soil. The disease status of tomato plants was assessed daily by scoring the disease index on a scale of 0 to 4 as previously described [35]. All plants were monitored for disease index analysis, and the following scale was used: 0, no symptoms; 1, 1–25% wilted leaves; 2, 26–50% wilted leaves; 3, 51–75% wilted leaves; and 4, 76–100% wilted leaves. *R. solanacearum* cells were grown in CPG medium to $OD_{600}$ = 1.0. Aliquots of 5 mL of the *R. solanacearum* WT, Δ*raaR*, and Δ*raaR*(raaR) CPG liquid medium were inoculated onto the tomato seedlings. Each treatment included 20 replicates [29,35,39].

## Analysis of *R. solanacearum* cell numbers in plants

One-gram samples of plant roots and stems were collected and milled in 9 mL of sterilized water for 20 min. The suspensions were serially diluted and spread on CPG plates. The plates were incubated at 28°C for 48 h. Then, the numbers of *R. solanacearum* cells were counted [29,39].

## Statistical analysis

Statistical analyses were performed with GraphPad Prism 8. The data are presented as the means ± standard deviations. Asterisks in figures indicate corresponding statistical significance as it follows: *, $p < 0.05$; **, $p < 0.01$; ***, $p < 0.001$.

## Supporting information

**S1 Fig. The effect of *trpEG* on the expression level of *raaR* was evaluated by assessing the β-galactosidase activity of *raaR-lacZ* transcriptional fusions in the wild-type, the *trpEG* mutant strains and the *trpEG* deletion mutant strain supplemented with anthranilic acid.** The data are the means ± standard deviations of three independent experiments. **$p < 0.01$; ***$p < 0.001$ (unpaired *t*-test).
(DOCX)

**S2 Fig.** Effects of *raaR* on the growth curve of *R. solanacearum* GMI1000 in CPG medium (A) and MP minimal medium (B). The cells were inoculated at 28°C in three replicates with low-intensity shaking in the Bioscreen-C automated growth curve analysis system. The experiment was started at initial $OD_{600}$ values of 0.01 in CPG medium and 0.1 in MP minimal medium. The data are the means ± standard deviations of three independent experiments.
(DOCX)

**S3 Fig. Influence of *raaR* on the expression level of housekeeping gene *gyrB*.** (A) Effect of *raaR* on the gene expression levels of *gyrB* in *R. solanacearum* GMI1000. We used promoter activity assays to quantify gene expression. (B) EMSA analysis of the *in vitro* binding of RaaR to the promoters of *gyrB*.
(DOCX)

**S4 Fig. ITC analysis of the binding between anthranilic acid and the LysR_substrate domain of the RaaR protein.** (A) SDS-PAGE of the purified LysR_substrate domain of RaaR protein. (B) ITC titration of 20 μM LysR_substrate domain of RaaR with 250 μM anthranilic acid in PBS buffer at 25°C.
(DOCX)

**S5 Fig. CD spectroscopy of anthranilic acid binding to RaaR protein.** The α-helix and β-sheet spectra of the RaaR protein were changed after RaaR protein was supplemented with anthranilic acid at a final concentration of 10 μM.
(DOCX)

**S6 Fig. The effects of HHQ on the binding of RaaR to the promoters of *phcB* (A) and *solI* (B) were assessed by performing EMSA *in vitro*.** A protein-DNA complex was formed when the protein was incubated with the probes, and different concentrations of HHQ showed no effect on the formation of the complex at room temperature for 30 min.
(DOCX)

**S7 Fig. Differential gene expression profiles between *R. solanacearum* wild-type strain GMI1000, *trpEG* and *raaR* mutants as measured by RNA-Seq (Log$_2$ fold-change ≥1.5).** (A)

GO term enrichment analysis of differentially expressed genes between the Δ*raaR* and wild-type strains. Venn diagrams showing the overlap of genes with (B) upregulated or (C) downregulated expression on different mutant backgrounds. Divergently regulated genes are not depicted in these Venn diagrams but are found in S2 Table.
(DOCX)

**S8 Fig.** Effects of *mvfR* of *P. aeruginosa* on the RaaR-regulated motility (A), biofilm formation (B), EPS production (C) and cellulase production (D) in the *R. solanacearum raaR* deletion mutant strain. The data are the means ± standard deviations of three independent experiments. $^{**}p < 0.01$; $^{***}p < 0.001$ (unpaired *t*-test).
(DOCX)

**S9 Fig. ITC analysis of the binding of PQS or anthranilic acid to MvfR protein.** (A) SDS-PAGE of the purified MvfR protein. (B) ITC titration of 20 μM MvfR protein with 250 μM PQS in PBS buffer at 25°C. (C) ITC titration of 20 μM MvfR with 250 μM anthranilic acid in PBS buffer at 25°C.
(DOCX)

**S10 Fig.** ITC analysis of the binding between (A) PQS, (B) HHQ and (C) DHQ and RaaR protein. ITC titration of 20 μM RaaR protein with 250 μM PQS, HHQ and DHQ in PBS buffer at 25°C.
(DOCX)

**S11 Fig. Schematic representation of the regulatory network of the anthranilic acid/RaaR signaling system in *R. solanacearum*.** RaaR is involved in sensing anthranilic acid signals and increases the expression levels of PhcB and SolI, which are required for the synthesis of 3-OH PAME/3-OH MAME and AHL signals, respectively. At the same time, the anthranilic acid/RaaR signaling system also directly controls the motility, biofilm formation, and virulence of *R. solanacearum*.
(DOCX)

**S12 Fig. Analysis of the binding of anthranilic acid to RaaR variants.** (A) SDS-PAGE of the purified RaaR$^{A171F}$, RaaR$^{L191A}$, RaaR$^{L192A}$ and RaaR$^{I249A}$ proteins. ITC analysis of the binding of anthranilic acid to (B) RaaR$^{A171F}$, (C) RaaR$^{L191A}$, (D) RaaR$^{L192A}$ and (E) RaaR$^{I249A}$ proteins.
(DOCX)

**S13 Fig. Influence of *raaR* on the *epsA* gene expression of *R. solanacearum*.** (A) Effects of *raaR* on the gene expression levels of *epsA* in *R. solanacearum* GMI1000. We used promoter activity assays to quantify gene expression. (B) EMSA analysis of the *in vitro* binding of RaaR to the promoter of *epsA*. The biotin-labeled 336-bp *epsA* promoter DNA probe was used for the protein binding assay. A protein-DNA complex, represented by a band shift, was formed when different concentrations of RaaR protein were incubated with the probe at room temperature for 30 min.
(DOCX)

**S14 Fig. Analysis of the RaaR binding sites in the promoter regions of target genes.** Analysis of the binding between RaaR and the mutated *phcB* (A), *solI* (B) and *epsA* (C) promoters with deletion of the RaaR binding sequence GCGGGTGCG. EMSA analysis was performed *in vitro*.
(DOCX)

**S1 Table. Analysis of the homologs of MvfR in *R. solanacearum*.**
(DOCX)

**S2 Table. List of genes differentially expressed in the *trpEG* and *raaR* mutants compared to the wild-type strain (Log$_2$-fold change $\geq$ 1.5).** Significantly differentially expressed genes were determined by Cufflinks after Benjamini-Hochberg correction. The fold-change is the ratio of the mutant FPKM to the wild-type FPKM.
(DOCX)

**S3 Table. Analysis of the homologs of RaaR in various bacteria.**
(DOCX)

**S4 Table. Bacterial strains and plasmids used in this study.**
(DOCX)

**S5 Table. PCR primers used in this study.**
(DOCX)

## Author Contributions

**Conceptualization:** Shihao Song, Yinyue Deng.

**Data curation:** Shihao Song, Xiuyun Sun, Quan Guo, Binbin Cui, Yu Zhu, Xia Li.

**Formal analysis:** Shihao Song, Jianuan Zhou, Lian-Hui Zhang, Yinyue Deng.

**Funding acquisition:** Yinyue Deng.

**Investigation:** Shihao Song, Xiuyun Sun, Quan Guo, Binbin Cui, Yu Zhu, Xia Li.

**Methodology:** Shihao Song.

**Project administration:** Shihao Song.

**Resources:** Shihao Song.

**Software:** Shihao Song.

**Supervision:** Yinyue Deng.

**Validation:** Shihao Song.

**Visualization:** Shihao Song.

**Writing – original draft:** Shihao Song, Yinyue Deng.

**Writing – review & editing:** Shihao Song, Yinyue Deng.

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
