## [Decision Letter · Decision Letter 0]

11 Jan 2022

Dear Dr. Deng,

Thank you very much for submitting your manuscript "A responsive transcriptional regulator controls Ralstonia solanacearum pathogenicity by sensing anthranilic acid" for consideration at PLOS Pathogens. As with all papers reviewed by the journal, your manuscript was reviewed by members of the editorial board and by several independent reviewers. In light of the reviews (below this email), we would like to invite the resubmission of a significantly-revised version that takes into account the reviewers' comments.

The three reviewers all commended the discovery of the novel sensor of anthranilic acid RaaR in Ralstonia solanacearum. The reviewers also identified some major issues that need to be addressed including to further corroborate the relationship between RaaR and anthranilic acid and EMSA experiments. Please also move the method section in the supplementary information to the main text. In addition, please conduct the experiment to restore ΔtrpEG with exogenous anthranilic acid if this has not been done in the past.

Please show images of infected and control plants corresponding to Fig. 6A in the supplementary

Please move Fig. S1 to the main text.

We cannot make any decision about publication until we have seen the revised manuscript and your response to the reviewers' comments. Your revised manuscript is also likely to be sent to reviewers for further evaluation.

Sincerely,

Nian Wang

Associate Editor

PLOS Pathogens

Wenbo Ma

Section Editor

PLOS Pathogens

Kasturi Haldar

Editor-in-Chief

PLOS Pathogens

orcid.org/0000-0001-5065-158X

Michael Malim

Editor-in-Chief

PLOS Pathogens

orcid.org/0000-0002-7699-2064

The three reviewers all commended the discovery of the novel sensor of anthranilic acid RaaR in Ralstonia solanacearum. The reviewers also identified some major issues that need to be addressed including to further corroborate the relationship between RaaR and anthranilic acid and EMSA experiments. Please also move the method section in the supplementary information to the main text. In addition, please conduct the experiment to restore ΔtrpEG with exogenous anthranilic acid if this has not been done in the past.

Please show images of infected and control plants corresponding to Fig. 6A in the supplementary

Please move Fig. S1 to the main text.

Reviewer's Responses to Questions

**Part I - Summary**

Reviewer #1: The manuscript titled "A responsive transcriptional regulator controls Ralstonia solanacearum pathogenicity" by Song et al. characterize RaaR, a novel transcriptional regulator that function upstream to the biosynthesis of Ralstonia solanacearum QS signals AHL and 3-OH MAME by directly controlling the expression of phcB and solI which results in reduction in exoenzyme activity, EPS production, motility and virulence. RaaR was identified to complement the anthranilic acid biosynthesis mutant trpGE and therefore was reported in the manuscript to act downstream to anthranilic acid.

The research subject is interesting, the manuscript is well written and the results are described in clear and organized manner. However, I do not think that the authors provided enough data to support that RaaR indeed function downstream and not in parallel to anthranilic acid and some key controls are missing in some of the experiments. Therefore, I suggests major revisions.

Reviewer #2: Summary of the Research.

This manuscript describes the discovery of the novel sensor of anthranilic acid RaaR in the economically important plant pathogen Ralstonia solanacearum (Rs). The authors build off a previous study which found that anthranilic acid can positively regulate a variety of important virulence traits in Rs, including motility, biofilm production, and the production of plant cell wall degrading enzymes. Based on structural similarity to the quorum sensing receptor MvfR in Pseudomonas aeruginosa, the authors studied the putative receptor RSp0912, which they named RaaR. Using both genetic and biochemical techniques, the authors demonstrate that RaaR is required for anthranilic acid signaling, physically binds both anthranilic acid and several key promoters, and contributes to Rs virulence. The authors end by proposing that RaaR and anthranilic acid contributes to the complex regulation of virulence factors in Rs. They further suggest that, because they can detect anthranilic acid in the tissues of some Rs hosts, anthranilic acid could be an interkingdom signal between host and pathogen.

Strengths

The authors admirably use a variety of means to show how RaaR interacts with anthranilic acid and gene expression. Their genetic methods use both RaaR mutants and RaaR overexpression constructs, reporters of putative RaaR-regulated genes, and complementation of RaaR mutants both with the gene itself and with exogenously applied anthranilic acid. They use EMSA, ITC, and CD to demonstrate that RaaR binds the promoters it regulates and to anthranilic acid. Combined, these multiple lines of evidence show very clearly and convincingly that RaaR functions as they suggest. They further make a successful effort to show the specificity of the RaaR-anthranilic acid interaction, and convincingly show that RaaR is not affected by other related QS molecules. Further, their result showing that virulence on tomato is lowered in an RaaR mutant shows that this is important for Rs, and that this regulatory pathway plays a significant role in an important plant pathogen.

Concerns

One area where this paper falls short is in examining the breadth of the R. solanacearum species complex. When they search for RaaR homologs in Table S2, it would be useful to show if RaaR is conserved across Rs phylotypes. They could include comparisons to strains like CFBP2957 (Phylotype II), UW551 (Phylotype II), CMR15 (Phylotype III), PSI07 (Phylotype VI), or other strains with available genome sequences. The authors also suggest in the discussion that because they can detect anthranilic acid in some Rs hosts but not in non-host maize or rice, anthranilic acid may help Rs with host sensing. However, the three hosts they test are all solanaceous crops (tomato, eggplant, and pepper) and do not capture the diversity of the Rs host range. Some individual strains, including GMI1000, can have extremely wide host ranges. It would be better to include a wider variety of Rs hosts in this test, including monocot hosts like banana and ginger. However, this experiment is not vital to the main focus of the manuscript and could be simply removed. If this supplemental figure is included, the authors should also provide more detail in how they sampled tissues from the tested crops.

Reviewer #3: In this manuscript “A responsive transcriptional regulator controls Ralstonia solanacearum pathogenicity by sensing anthranilic acid” (PPATHOGENS-D-21-02479) the authors identified a new signaling system formed by one-component system RaaR (LysR-type regulator) and signal molecule anthranilic acid. RaaR-anthranilic acid complex was verified by ITC and CD experiments. Also, it was suggested that anthranilic acid binding caused conformational changes in RaaR, which positively affected its interaction with target promoters. I have just a few suggestions to reinforce these interesting findings and clarify some parts in text.

**Part II – Major Issues: Key Experiments Required for Acceptance**

Reviewer #1: 1. The major weakness of the paper is that it does not provide sufficient proof that RaaR act downstream to anthranilic acid and not in parallel to it. If RaaR regulate QS signal synthesis, its overexpression might rescue the trpGE phenotypes regardless if it act downstream to it or not. To clearly demonstrate that RaaR and RrpGE function in the same pathway, epistasis experiments in the presence and absence of anthranilic acid are required:

a. The authors should produce a raaR/trpGE double mutant and examine if their phenotypes (phcB and solI expression by LacZ promter fusions, motility, exoenzyme activity) are additive or not.

b. The authors should try to rescue these phenotypes by exogenous addition of anthranilic acid (as conducted in fig 6 in authors ISME paper): If indeed RaaR act downstream to TrpGE, exogenous addition of anthranilic acid will rescue the trpGE mutant but not the trpGE/raaR double mutant.

2. Some of the experiments described in the manuscript are lacking proper controls:

a. The authors demonstrated that RaaR binds to the phcB, solI and espA promoters in vitro by EMSA. Many transcriptional regulators harbors general affinity to DNA in in vitro, which results in high false positive rate in EMSA analysis when conducted without proper controls. As far as I see, all EMSA experiments used with RaaR in this study provided positive results, which make one wonder how specific is the binding of RaaR to these particular promoters. To address this issue the authors should repeat the EMSA experiments using promoters of genes that are NOT directly regulated by RaaR and show that RaaR demonstrate high affinity to phcB, solI and espA promoters and not others.

b. To make sure that RaaR does not harbor a pleiotropic effect on protein expression or LacZ activity, the authors should use a negative control (maybe a promotor of a housekeeping gene fused to LacZ) in their promoter LacZ fusion assays.

c. Figure 5. The authors should conduct their binding assays with one of the structural analogs of anthranilic acid that were used in fig S8 as a negative control.

Reviewer #2: The authors should comment on the presence and conservation of RaaR within the R. solanacearum species complex.

Reviewer #3: Major points:

1) As anthranilic acid is produced from chorismic acid and L-glutamine and this pathway is necessary to synthesis of L-tryptophan in microorganisms, it is easy to think that anthranilic acid is essential to bacterial survive in condition without tryptophan in the growth medium. Also, it seems that most bacterial species can synthesize anthranilic acid but they do not encode a RaaR homologs in their genomes (only a few groups of bacterial species that carry a raar homolog were cited in the text). In this case, most bacterial specie can synthesize anthranilic acid but may not encode for a RaaR homolog but still can communicate with other species that carries a raar homolog in their genomes (cross-talking). What the authors think about this point? Can it be better discussed in the manuscript?

2) In material and methods are lacking description of RaaR expression in E. coli and purification, ITC and CD experiments including concentration of purified protein and ligand as well as instruments that were used in these studies. Also, how the QS signal molecules were isolated and quantified and the disease Index was calculated in virulence assays. Please, add more details about these experiments in Material and Methods.

3) It will be important the authors calculate and define the interaction parameter “Kd” for the RaaR-anthranilic acid complex. This result can help in the discussion about physiological concentration of anthranilic acid in bacterial cell and in host root/stem by affecting RaaR activation.

4) Have the authors identified any conserved cis-element in the RaaR target promoters? If so, it will be important to generate site mutants on this putative motif to confirm the RaaR specific binding by in vitro experiments.

**Part III – Minor Issues: Editorial and Data Presentation Modifications**

Reviewer #1: 1. The data provided in figure S2 suggest that RaaR is regulating its own transcription. Have the authors looked into this matter?

2. It will be nice if the virulence test conducted in Fig 6 will be supplemented with representative pictures.

3. Have the authors tested the expression or promoter activity of epsA in the raaR mutant? If not, what is the point of doing EMSA?

4. As far as I know PLOS pathogens don’t have word limitation. Therefore, it is unclear to me why the materials and methods section was split between the main text and the supplementary material.

Reviewer #2: -Line 65 – There are two more recent studies of phc regulation that may be worth citing here: Perrier et al. (2018), Microbial Pathogenesis (doi: 10.1016/j.micpath.2018.01.028) and Khokhani et al. (2017), mBio (doi: 10.1128/mBio.00895-17)

-Line 273 – The authors should clarify what they mean by “fermentation broth”. If it is not CPG or a water suspension, the authors should specify the components of fermentation broth

-Line 426 – Did the authors perform any statistical tests on the disease progress data in figure 6A? Possible tests to perform might include repeated measures ANOVA or AUDPC (see Schandry (2017), Front Plant Sci (doi: 10.3389/fpls.2017.00623))

-Supplementary Table 2 – The authors include in this table R. solanacearum GMI1000 and R. pseudosolanacearum RS 476. If the authors are using the division of the Ralstonia solanacearum species complex (RSSC) into three species (Prior et al. (2016), BMC Genomics (doi: 10.1186/s12864-016-2413-z)), GMI1000 should also be referred to as R. pseudosolanacearum, as it is also a Phylotype I strain. Alternatively, the authors should refer to RS 476 as R. solanacearum and include its Phylotype. The authors should also include a line in the introduction describing the RSSC and refer to GMI1000 throughout as R. pseudosolanacearum.

Reviewer #3: Minor points:

Line 29: RaaR regulates the same processes as anthranilic acid, and both are conserved in various bacterial species. In this sentence should be better to say that anthranilic acid is present and not conserved in bacterial species.

Line 30: Please, add ”a” before “anthranilic acid-deficient mutant phenotypes were rescued by in trans expression of RaaR” and explain in the text way a trans expression can recue mutant phenotypes without adding anthranilic acid. It will help to clarify this part for the readers.

Line 68: “The signals phosphorylate the response regulator PhcR by activating the histidine kinase” should be “The signals promote the phosphorylation of the response regulator PhcR

by…”

Line 128: Please, indicate the percentage of recue for the sentences in text that are described “almost fully rescued signal”.

PLOS authors have the option to publish the peer review history of their article (what does this mean?). If published, this will include your full peer review and any attached files.

Reviewer #1: No

Reviewer #2: No

Reviewer #3: No
---

## [Decision Letter · Decision Letter 1]

30 Mar 2022

Dear Dr. Deng,

Thank you very much for submitting your manuscript "An anthranilic acid-responsive transcriptional regulator controls the physiology and pathogenicity of Ralstonia solanacearum" for consideration at PLOS Pathogens. As with all papers reviewed by the journal, your manuscript was reviewed by members of the editorial board and by several independent reviewers. The reviewers commended the revisions and extra data as requested and appreciated the attention to an important topic. Based on the reviews, we are likely to accept this manuscript for publication, providing that you modify the manuscript according to the review recommendations.

Sincerely,

Nian Wang

Associate Editor

PLOS Pathogens

Wenbo Ma

Section Editor

PLOS Pathogens

Kasturi Haldar

Editor-in-Chief

PLOS Pathogens

orcid.org/0000-0001-5065-158X

Michael Malim

Editor-in-Chief

PLOS Pathogens

orcid.org/0000-0002-7699-2064

Reviewer Comments (if any, and for reference):

Reviewer's Responses to Questions

**Part I - Summary**

Reviewer #1: The authors have successfully addressed all the issues I raised in the first review cycle. The manuscript is well written and the data is solid and presented in a clear and organized manner. I suggest some minor text revisions that will improve the paper. Outside of that, I do not have any additional requests.

Reviewer #3: The authors have answered the reviewer´s concerns and added new data which reinforce their findings. The authors described changes in RaaR secondary structure caused by anthranilic acid. Although, I think the binding of anthranilic acid and RaaR should be better explored by using some mutant in critical residues of RaaR that are involved in this interaction. The lost of this interaction should be observed in ITC experiments.

**Part II – Major Issues: Key Experiments Required for Acceptance**

Reviewer #1: None

Reviewer #3: Identification of the RaaR critical residues involved in interaction with anthranilic acid should be accomplished to give the reader a complete understanding of this mechanism. Mutant in critical residues of RaaR generated by site direct mutagenesis may help to answer this important question.

**Part III – Minor Issues: Editorial and Data Presentation Modifications**

Reviewer #1: Suggested minor text modifications (lines numbers in "track changes" draft):

*promoter fusion assays: throughout the manuscript, the authors used the term "gene expression" to describe their promoter fusions assays. In most cases, the term "gene expression" is referred to RNA accumulation (usually by RT-qPCR) and not reporter fusions analyses. The authors should clearly state in the text they used promoter activity assays to quantify gene expression.

*Line 118: a sentence should not start with "and"

*Line 203, 205,207, and 209: trpEG and raaR should be italicized

*Line 228: all the mentioned bacterial genera belong to the Burkholderiales order. This should be stated in the text.

*lines 260-265/Fig S13- The data presented here is more suited to be part of the result section and not the discussion section

*line 294-303 (Protein Expression and Purification) - MvfR was cloned to an expression vector and the protein was purified as well. This should be added to the M&M section.

*line 325 – citation is missing after "as described previously"

*Figure 1: the title "Overexpression of the trpEG mutant with raaR" is not clear. The authors should consider rephrasing it.

*Figure 1A: " Effects of the homologs…". Does not describe the presented experiment well. The authors should mention that the regulators are overexpressed and that the bacteria were assayed for plate motility.

Figures 3A and B/S3A/S11A: the authors should state that experiments present promoter activity analyses.

Figure 6A: the authors should state in the figure legend at which time after inoculation these pictures were taken.

Figure S7: ∆raaR should be italicized

Reviewer #3: (No Response)

PLOS authors have the option to publish the peer review history of their article (what does this mean?). If published, this will include your full peer review and any attached files.

Reviewer #1: No

Reviewer #3: No

Figure Files:

Data Requirements:

Reproducibility:

References:

---

## [Editor Report · Decision Letter 2]

29 Apr 2022

Dear Dr. Deng,

We are pleased to inform you that your manuscript 'An anthranilic acid-responsive transcriptional regulator controls the physiology and pathogenicity of Ralstonia solanacearum' has been provisionally accepted for publication in PLOS Pathogens.

Please address some minor editorial issues during the proof stage.

For example: 

Line 254. please add that before might. 

Analysis of the two proteins identified four conserved amino acid residues at positions A171, L191, L192 and I249 that might be critical for the interaction between RaaR and anthranilic acid.

Line 312: 312 The coding regions 

Line 315: should be was

Here the line numbers refer to the document tracking changes. 

Best regards,

Nian Wang

Associate Editor

PLOS Pathogens

Wenbo Ma

Section Editor

PLOS Pathogens

Kasturi Haldar

Editor-in-Chief

PLOS Pathogens

orcid.org/0000-0001-5065-158X

Michael Malim

Editor-in-Chief

PLOS Pathogens

orcid.org/0000-0002-7699-2064
---

## [Editor Report · Acceptance letter]

23 May 2022

Dear Dr. Deng,

We are delighted to inform you that your manuscript, " An anthranilic acid-responsive transcriptional regulator controls the physiology and pathogenicity of Ralstonia solanacearum ," has been formally accepted for publication in PLOS Pathogens.

Best regards,

Kasturi Haldar

Editor-in-Chief

PLOS Pathogens

orcid.org/0000-0001-5065-158X

Michael Malim

Editor-in-Chief

PLOS Pathogens

orcid.org/0000-0002-7699-2064